# Dyadic and Individual Variation in 24-Hour Heart Rates of Cancer Patients and Their Caregivers

**DOI:** 10.3390/bioengineering11010095

**Published:** 2024-01-18

**Authors:** Rajnish Kumar, Junhan Fu, Bengie L. Ortiz, Xiao Cao, Kerby Shedden, Sung Won Choi

**Affiliations:** 1Department of Pediatrics, University of Michigan Medical School, Ann Arbor, MI 48109, USA; rajnishk@umich.edu (R.K.); bengielo@umich.edu (B.L.O.); caxiao@umich.edu (X.C.); 2Department of Statistics, College of Literature, Science, and the Arts, University of Michigan, Ann Arbor, MI 48109, USA; jasonfu@umich.edu (J.F.); kshedden@umich.edu (K.S.)

**Keywords:** heart rate (HR), cancer, hematopoietic cell transplantation (HCT), dyad, circadian, canonical correlation analysis, wearable monitor

## Abstract

Background: Twenty-four-hour heart rate (HR) integrates multiple physiological and psychological systems related to health and well-being, and can be continuously monitored in high temporal resolution over several days with wearable HR monitors. Using HR data from two independent datasets of cancer patients and their caregivers, we aimed to identify dyadic and individual patterns of 24 h HR variation and assess their relationship to demographic, environmental, psychological, and clinical variables of interest. Methods: a novel regularized approach to high-dimensional canonical correlation analysis (CCA) was used to identify factors reflecting dyadic and individual variation in the 24 h (circadian) HR trajectories of 430 people in 215 dyads, then regression analysis was used to relate these patterns to explanatory variables. Results: Four distinct factors of dyadic covariation in circadian HR were found, contributing approximately 7% to overall circadian HR variation. These factors, along with non-dyadic factors reflecting individual variation exhibited diverse and statistically robust patterns of association with explanatory variables of interest. Conclusions: Both dyadic and individual anomalies are present in the 24 h HR patterns of cancer patients and their caregivers. These patterns are largely synchronous, and their presence robustly associates with multiple explanatory variables. One notable finding is that higher mood scores in cancer patients correspond to an earlier HR nadir in the morning and higher HR during the afternoon.

## 1. Introduction

A dyad is a social unit composed of two individuals. The dynamics within this unit are influenced by both independent and interdependent interactions embedded within broader social contexts. Independent effects refer to the individual experiences of each member of the dyad, while interdependent effects reflect the actions, behaviors, or circumstances involving one member of the dyad and are subsequently influenced by the other [1]. Accordingly, dyadic data, including information gathered from relationships, such as romantic partners, patient–caregiver pairs, or therapist–client pairs, offer rich insights into these key relational dynamics [2].

One key illustration can be found in the context of cancer. When a patient is diagnosed with cancer, it impacts the entire family [3,4]. Often, family members step into the role of informal caregivers, exposing them to physical and psychological strain over long periods of time [5]. Not surprisingly, the distress experienced by both patients and caregivers interplay with each other [6]. Indeed, research suggests that depressed mood and poor health-related quality of life in caregivers may elevate the risk of depressive symptoms in patients with cancer [7]. However, the analyses of such phenomena present their own challenges [4]. These dyadic interactions ideally are captured longitudinally [8], and any dyadic variation may be obscured by stronger sources of individual variation.

Previously, research on dyads relied on self-reported data (i.e., patient-reported outcomes, PRO), often from only with one member of a dyad [9]. In recent years, technological advances have enabled the collection of multi-parameter physiologic data using consumer-grade devices at low cost in real-time [10]. However, there have not yet been sufficient clinical studies to establish what parameters should be measured, at what frequency, and with what level of data quality [11,12,13]. One promising parameter is heart rate (HR), which historically has been viewed as an output of emotional reactions or stress. Indeed, the 24 h HR trajectory has been cited as a potential biomarker of physiological and psychological health [14].

Digitally derived biomarker data from individuals is now possible with smartphone apps and wearables that can collect longitudinal data [15]. Advances in data science and computing power support the analysis and interpretation of such “big data” [16]. These advances led our team to design an mHealth app called *Roadmap*, which integrates data from non-invasive wearable sensors (Fitbit) and self-reported mood to monitor mental health. Initially developed to provide resilience-enhancing support to patient–caregiver dyads undergoing hematopoietic cell transplantation (HCT), the app was also adapted for other patient–caregiver dyads with cancer (“ONC”) [17,18,19].

In this paper, we aimed to identify 24 h dyadic patterns by analyzing HR data from wearable Fitbit sensors gathered from two recently completed mHealth studies. We devised and applied a dimensionality reduction approach that revealed complex variations in 24 h HR trajectories. We specifically sought factors that were maximally correlated between patients and caregivers, leading us to identify dyadic covariation in overall HR variation. We then explored a set of individual, dyadic, and environmental explanatory variables, and established relationships between these variables and the contribution of each factor to the HR trajectory on a given person/day.

## 2. Materials and Methods

### 2.1. Description of Wearable Devices

After providing Institutional Review Board (IRB)-approved informed consent, subjects were asked to wear a Fitbit Charge smartwatch. These devices are advanced touch-screen fitness trackers that monitor variables including HR. The device can be worn on either wrist and wirelessly connects with the subject’s smartphone via Bluetooth to the Fitbit mobile app, available on Android and iOS systems. Subjects had access to their own data. The subject was instructed to wear the Fitbit device continuously (except for charging or showering) over a 120-day study period. An overview of data collection is shown in Figure 1. The smartwatch device uploaded data to the subject’s phone every 15 min via Bluetooth Low Energy. The study team had access to the subjects’ dashboards and data download for analysis through the Roadmap 2.0 app.

### 2.2. Data Preprocessing

Analysis was restricted to the initial consecutive 120 days of data for each subject. For each 5 min interval on a specific day, the median HR for each subject was calculated, with a missing value indicator entered when no HR measurements were available within the interval. For each subject/day, we therefore have up to 120 vectors of dimension 288, containing 5 min median HR values. Patient/caregiver dyads were linked, giving rise to up to 120 dyad-day vectors of length 288×2=576. More formally, let $H_{dtm}^{p}$ and $H_{dtm}^{c}$ denote the patient and caregiver HRs for dyad d=1,…,215 on day t=1,…,120 and 5 min block m=1,…,288, and let $W_{dtm}^{p}$ and $W_{dtm}^{c}$ denote the corresponding binary masks such that $W_{dtm}^{*}=1$ if the HR was observed for index (d,t,m) and $W_{dtm}^{*}=0$, otherwise with ∗∈{p,c} for patients and caregivers.

### 2.3. Factor Identification

#### 2.3.1. Estimation of Means and Covariances

For each of the 288 5 min intervals within a day, we estimated the marginal mean HR for patients and for caregivers using the sample mean of all available patient × day data. This results in trajectory vectors $\mu_{p},\mu_{c}\in R^{288}$. Specifically, the patient marginal mean at minute *m* is
(1)$$\mu_{m}^{p}=\sum_{dt}H_{dtm}^{p}W_{dtm}^{p}/\sum_{dt}W_{dtm}^{p},$$
and analogously for caregivers. We note that this is the marginal mean of all person days for a given dyadic role (patient or caregiver). Our focus is primarily on the individual person-day and dyad-day deviations from these means, which we also refer to as *anomalies*.

The observed patient × day HR trajectories were then centered with respect to $\mu_{p},\mu_{c}$, i.e., we set ${\tilde{H}}_{dtm}^{*}=H_{dtm}^{*}-\mu_{m}^{*}$ when $H_{dtm}^{*}$ is observed, and otherwise set ${\tilde{H}}_{dtm}^{*}$ to be missing. Then, the joint 576×576 patient × caregiver covariance matrix *S* was estimated, using all available data to estimate each element $S_{ij}$ using the usual product-moment estimator:(2)$$\begin{array}{ccc} S_{ij} & = & \sum_{dt}{\tilde{H}}_{dti}^{p}{\tilde{H}}_{dtj}^{p}W_{dti}^{p}W_{dtj}^{p}/\sum_{dt}W_{dti}^{p}W_{dtj}^{p}\;\;\left(i,j\leq 288\right) \end{array}$$(3)$$\begin{array}{ccc} S_{ij} & = & \sum_{dt}{\tilde{H}}_{dti}^{p}{\tilde{H}}_{dtj}^{c}W_{dti}^{p}W_{dtj}^{c}/\sum_{dt}W_{dti}^{p}W_{dtj}^{c}\;\;\left(i\leq 288,j>288\right) \end{array}$$(4)$$\begin{array}{ccc} S_{ij} & = & \sum_{dt}{\tilde{H}}_{dti}^{c}{\tilde{H}}_{dtj}^{c}W_{dti}^{c}W_{dtj}^{c}/\sum_{dt}W_{dti}^{c}W_{dtj}^{c}\;\;\left(i,j>288\right) \end{array}$$

The symmetric matrix *S* was then mapped into the cone of positive-semidefinite symmetric matrices by writing $S=BDB^{\prime }$ using the spectral decomposition and soft thresholding the eigenvalues in *D* at $\epsilon =10^{-4}$, yielding ${\tilde{D}}_{i}=\max\left(0,D_{i}-\epsilon \right)$, before reconstructing $\tilde{S}$ as $B\tilde{D}B^{\prime }$. Covariance matrices $C_{p}$ and $C_{c}$ for patients and for caregivers, and the cross-covariance matrix $C_{pc}$ between patients and caregivers, were then extracted from the appropriate blocks of $\tilde{S}=\left[C_{p}\;C_{pc};C_{pc}^{\prime }\;C_{c}\right]$. The matrices $C_{p}$, $C_{c}$, and $C_{pc}$ are all 288×288 in size.

#### 2.3.2. Objective Function Definition

To linearly reduce the dimension for interpretation, we aimed to identify the strongest dyadic correlation while preserving individual (subject-level) explained variation. Let $Q_{p}$ and $Q_{c}$ denote 288×q orthogonal matrices defining the (linear) dimension reductions for patients and caregivers, respectively. The columns of $Q_{p}$ and $Q_{c}$ are termed *loading vectors*. The dyadic correlation is captured by
(5)$$f\left(Q_{p},Q_{c}\right)=\mathrm{logdet}\left(Q_{p}^{\prime }C_{pc}Q_{c}\right)-\left(\mathrm{logdet}\left(Q_{p}^{\prime }C_{p}Q_{p}\right)+\mathrm{logdet}\left(Q_{c}^{\prime }C_{c}Q_{c}\right)\right)/2,$$
where logdet is the natural logarithm of the matrix determinant. This is similar to the canonical correlation analysis (CCA) objective function [20]. To preserve marginal variance we also consider the objectives
(6)$$\begin{array}{ccc} g_{p}\left(Q_{p}\right) & = & \mathrm{logdet}\left(Q_{p}^{\prime }C_{p}Q_{p}\right)-\mathrm{logdet}\left(Q_{p}^{\prime }Q_{p}\right)\\ g_{c}\left(Q_{c}\right) & = & \mathrm{logdet}\left(Q_{c}^{\prime }C_{c}Q_{c}\right)-\mathrm{logdet}\left(Q_{c}^{\prime }Q_{c}\right) \end{array}$$
which are similar to the principal component analysis (PCA) objective function [21]. The overall objective to maximize is
(7)$$f\left(Q_{p},Q_{c}\right)+\lambda_{p}g_{p}\left(Q_{p}\right)+\lambda_{c}g_{c}\left(Q_{c}\right)$$
where $\lambda_{p},\lambda_{c}>0$ are tuning parameters both set to 1/2 here.

#### 2.3.3. Objective Function Optimization

The objective function as defined above can be optimized using gradient descent, with the alternating directions method of multipliers (ADMM) [22] employed to impose the orthogonality constraints $Q_{p}^{\prime }Q_{p}=Q_{c}^{\prime }Q_{c}=I$, since the parameters naturally lie on the Stiefel manifold of *q*-frames in $R^{288}$, where *q* is the number of factors.

The objective function *f* (Equation 7) is invariant to the choice of orthogonal bases, i.e., for any q×q orthogonal matrices $R_{p}$ and $R_{c}$, $f\left(Q_{p},Q_{c}\right)=f\left(Q_{p}R_{p},Q_{c}R_{c}\right)$. In any factor-type analysis, to make the parameters identified, it is conventional to rotate the solution into some sort of canonical form. When correlations are of interest, it is natural to prefer a representation in which the scores are uncorrelated within and between the two blocks of variables being correlated (patient and caregiver HR trajectories, in our case). Therefore, following optimization, the basis matrices were rotated by constructing the unique r×r orthogonal matrices $R_{p}$ and $R_{c}$, such that ${\left(Q_{p}R_{p}\right)}^{\prime }C_{pc}\left(Q_{c}R_{c}\right)$ is a diagonal matrix with non-increasing diagonal values, both ${\left(Q_{p}R_{p}\right)}^{\prime }C_{p}\left(Q_{p}R_{p}\right)$ and ${\left(Q_{c}R_{c}\right)}^{\prime }C_{c}\left(Q_{c}R_{c}\right)$ are diagonal, and all columns of $Q_{p}R_{p}$ and $Q_{c}R_{c}$ are unit vectors. We then replaced $Q_{p}$ and $Q_{c}$ with $Q_{p}R_{p}$ and $Q_{c}R_{c}$, respectively.

### 2.4. Goodness of Fit of the Estimated Factors

We considered several metrics for quantifying the goodness of fit (explained variation) in the estimated factors. Let $C_{*}$ denote the p×p covariance matrix (for patients or for caregivers). For a unit loading vector $v\in R^{p}$, the ratio ${\mathrm{PVE}}_{\max}\equiv v^{\prime }C_{*}v/\lambda_{\max}\left(C_{*}\right)$ is the variance explained by *v* relative to the maximum possible variance explained by any other loading vector ($\lambda_{\max}\left(A\right)$ is the maximum eigenvalue of the positive semidefinite matrix *A*). Next, to quantify the variance explained by factors $V\in R^{p\times q}$ relative to the total variance, we first reduce the dimension from *p* to $p^{\prime }$ so that 80% of the total variance is preserved. This produces a new $p^{\prime }\times p^{\prime }$ covariance matrix ${\tilde{C}}_{*}$ and a new set of loadings $\tilde{V}$. Then, we construct $\tilde{M}=\tilde{V}{\tilde{V}}^{\prime }{\tilde{C}}_{*}\tilde{V}{\tilde{V}}^{\prime }$ and $\tilde{E}={\tilde{C}}_{*}-\tilde{M}$. Let $\left\{\lambda_{i}\right\}$ denote the eigenvalues of ${\tilde{E}}^{-1/2}\tilde{M}{\tilde{E}}^{-1/2}$, using the pseudo-inverse when $\tilde{E}$ is singular. The $\lambda_{i}$ can be interpreted as signal-to-noise ratios along elements of an orthogonal basis. For each eigenvalue $\lambda_{i}$, the ratio $\lambda_{i}/\left(1+\lambda_{i}\right)$ estimates the proportion of variance explained when projecting in the direction of the $i^{th}$ eigenvector. Pillai’s trace is the sum of these values, so we use the mean (Pillai’s trace divided by $p^{\prime }$) to summarize the explained variation over the dominant subspace of $C_{*}$ capturing 80% of the total variance, and refer to this as the “Pillai PVE”.

#### 2.4.1. Inference of Factor Scores

Factor scores were inferred using conditional expectations for the Gaussian distribution. Let $Y\in R^{288}$ denote a patient/day HR observation, let *I* denote the indices in *Y* that are observed, and let J=[1,…,288] denote all indices in *Y*. The factor scores for this observation were inferred as
(8)$$\begin{array}{c} s=Q_{p}^{\prime }E\left[Y|Y\left[I\right]\right]=Q_{p}^{\prime }C_{p}\left[J,I\right]C_{p}{\left[I,I\right]}^{-1}\left(Y\left[I\right]-\mu_{p}\left[I\right]\right), \end{array}$$
with the analogous expression used for caregivers.

#### 2.4.2. Assessing Dyadic Correlations for Statistical Significance

We used a split-sample approach to assess the statistical significance of dyadic correlations. In 100 independent rounds, we used 80% of the dyads to estimate the factor structure $\left(Q_{p},Q_{c}\right)$, and the correlations were then inferred based on covariance matrices ${\tilde{C}}_{pc}$, ${\tilde{C}}_{p}$, and ${\tilde{C}}_{c}$, calculated from the held-out 20% of dyads. This approach yields unbiased point estimates and 95% confidence intervals for the dyadic correlations.

An important decision in any factor-decomposition approach is the number of factors to retain. Here, we elected to include all factors with statistically significant dyadic variance (based on the 95% confidence intervals from the split-sample approach described above excluding zero), and also retain one additional factor to reflect the most prominent non-dyadic variance.

#### 2.4.3. Association of Factor Scores with Explanatory Variables

Each identified patient or caregiver HR factor corresponds to a distinct variance component that is uncorrelated with all other factors. To aid in the interpretation of these factors, linear regression models were used to assess the manner in which the variation in the scores for each factor is explained by variables of interest. These models were fit using generalized estimating equations (GEE) [23] to account for within-subject correlations across days, and within-dyad correlations within and among days. Factor scores constructed as discussed above were the dependent variables of these regression models, and various combinations of explanatory variables were considered. A separate model was fit for each factor. Mean structure was estimated using ordinary least squares (i.e., GEE with working independence correlation models), and robust variance estimates were obtained by clustering observations in the same dyad.

To aid in interpreting the results, we constructed biplots that represent all regression coefficients for all factors in a single plot. Suppose we extract *q* factors and use *p* explanatory variables in the regression analysis. To construct a biplot, let $C\in R^{p\times q}$ denote the matrix of standardized regression coefficients (a standardized regression coefficient ${\hat{\beta }}_{s}$ for a variable *X* is defined as ${\hat{\beta }}_{s}=\hat{\beta }\cdot \mathrm{SD}\left(X\right)$, where $\hat{\beta }$ is the unstandardized regression coefficient). We then decompose $C=USV^{\prime }$ using the singular value decomposition. The *variable scores* for variable *j* are $\left(S_{1}^{1/2}U_{j1},S_{2}^{1/2}U_{j2}\right)$ and the *factor scores* for factor *k* are $\left(S_{1}^{1/2}V_{k1},S_{2}^{1/2}V_{k2}\right)$. These scores can be plotted on the same axis to represent all regression coefficients in all factors.

While we aimed to interpret each factor, we can also use the combined contributions of all factors to estimate the partial effect, or predicted marginal mean for each explanatory variable of interest. Specifically, let $s_{j}^{*}$ denote the scores for factor j=1,2,…, with ∗ denoting patients or caregivers, and let ${\hat{s}}_{j}^{*}\left(k,\ell \right)$ denote the predicted value for $s_{j}^{*}$ using the models discussed above, for level *ℓ* of explanatory variable *k* while holding all other explanatory variables fixed at reference levels. The predicted circadian HR trajectory $\sum_{j}{\hat{s}}_{j}^{*}\left(k,\ell \right)Q_{*}\left(j\right)$ can be plotted for levels (ℓ) of variables (k), for patients or caregivers (∗). This approach to the analysis uses the factor structure to denoise and smooth the data, but does not aim to ascribe mechanistic roles to the different factors.

## 3. Results

Data were available for 215 dyads on 15,124 dyad days. For the 5 min resolution HR summaries, 58% of patient values were observed and 76% of caregiver values were observed. Figure 2 shows the mean HR trajectories over all 15,124 patient days and over all 19,036 caregiver days.

### 3.1. Anomalies in 24-Hour Heart Rate Trajectories

The focus of this analysis is on individual deviations from the patient or caregiver mean trajectories. The most common individual deviations are captured through factors, with the loading patterns for the top five factors shown in Figure 3. The black curve in each plot shows the mean HR trajectory and the red/blue curves show the mean plus or minus loading patterns corresponding to one score standard deviation.

Four factors showed substantial and statistically significant associations between patients and caregivers (Table 1). These factors cumulatively explain 7% and 6% of the HR variation in patients and caregivers, respectively. A fifth factor lacks evidence for dyadic association, but adds substantially to the overall proportion of explained variance. Since our focus is on dyadic HR variation, we do not explore additional non-dyadic factors.

Loading patterns for all factors are shown in Figure 3. These loading patterns reflect circadian anomalies that represent all discernible dyadic variation in the HRs, and that cumulatively explain 11% of variation in circadian HR patterns. Notably, patient and caregiver loading patterns within a factor are highly similar—this similarity emerges entirely from the data, as there is no such constraint imposed by the analysis approach. This implies that dyadic HR anomalies correspond primarily to synchronous patterns of deviation from the mean HR trajectory in patients and in caregivers.

The most prominent variation in the 24 h HR pattern is the timing of the HR nadir, which in the mean trajectory occurs at around 5 am for both patients and caregivers (Figure 2). Additional prominent variations are the timing and rate of the morning HR rise and the presence of a local peak between 9 am and noon. Variation in these characteristics is reflected in several of the five factors (Figure 3). In factors 1, 4, and 5, both patients and caregivers with higher scores have an earlier nadir compared to people with a lower score, and in all three factors this earlier nadir is followed by a peak that occurs at around 9 am for people with higher scores (red curves), versus at around noon for people with lower scores (blue curves).

Factors 2, 3, and 4 differentiate dyad days with a unimodal HR pattern from those with a bimodal HR pattern. Positive scores on factors 2 and 4 and negative scores on factor 3 correspond to bimodal patterns, while negative scores on factors 2 and 4 (caregivers only) and positive scores on factor 3 correspond to unimodal patterns. Unimodal peaks occur before noon on factors 2 and 4 and at around 3 pm on factor 3. When bimodal peaks occur, the first is at around 9 am and the second is at 6 pm (factor 2), 8 pm (factor 3), and 4 pm (factor 4). Both positive and negative deviations on factor 5 exhibit bimodal patterns, with positive deviations exhibiting peaks at 9 am and 4 pm, while negative deviations exhibit a second peak at 9 pm for patients. Bimodal patterns, when present, arise due to a local minimum in the HR that occurs at around 11 am (factor 2), 3 pm (factor 3), and 4 pm 9 (factor 5). In all such cases, these minima constitute a second HR nadir in addition to the more prominent nadir at around 7–8 am.

Several of the factors capture differences in the overnight HR levels. Patients with high scores on factors 1, 4, and 5 all have lower overnight HR, as do caregivers with high scores on factors 1 and 4. Both patients and caregivers with low scores on factors 2 and 3 have lower overnight HR, especially near the nadir.

### 3.2. Association between HR Circadian Rhythm and Explanatory Variables

Regression analysis was used to relate explanatory variables to scores on the factors discussed above. The explanatory variables can be characterized as individual (age, sex, mood), dyadic (relationship, treatment arm), or environmental (day of the week, season, day of treatment). Since mood was not reported on a substantial fraction of days, we first fit a model excluding mood, then fit a second model with a smaller number of observations including mood.

#### 3.2.1. Associations between Individual Circadian Factors and Explanatory Variables

Table 2 shows Z-scores for the main effects of all covariates as predictors of each factor in patients and in caregivers. Z-scores exceeding a nominal significance threshold of 2 are shown in bold. All variables except for the caregiver relationship are statistically significant in at least one factor, and all factors have multiple significant predictors. Variables showing strong associations (|Z|≥3) with at least one factor are: sex, age, mood, days of week, and treatment day.

Biplots (Figure 4) aid in interpreting the regression analyses, by displaying the common and distinct relationships between predictor variables and the scores for the five factors of interest. These relationships are presented graphically in two-dimensional representations. An axis, plotted as a gray line, is constructed for each of the five factors under consideration. All biplots depicted explain at least 85% of the variation among regression effects. The longest axes (gray lines) in the biplots reflect the covariate effects that are strongest and best captured by the biplot. Axes that are approximately parallel correspond to common patterns of covariate effects in several factors, while axes that are approximately perpendicular correspond to distinct patterns of covariate among several factors. The longest and arguably most important axes in the biplot correspond to factors 2, 4, and 5. The axes corresponding to factors 2 and 4 are nearly parallel, but are oriented in opposing directions. This indicates that the same variables predict the scores for these two factors, but with coefficients having opposing signs, which is consistent with Table 2. The axis for factor 5 is approximately orthogonal to those for factors 2 and 4, indicating that these two sets of factors have distinct patterns of association with covariates. Factor 1 is nearly intermediate to factors 2/4 and 5 in terms of the predictor variables patterns revealed in the biplot. Since the point for factor 3 falls close to the origin, the biplot does not reflect the predictors of factor 3 well and, indeed, based on Table 1, there are few discernible predictors of factor 3.

The loading patterns for factors 2 and 4 (Figure 3) are similar except for the overnight HR levels. Combining this observation with the previously noted opposing patterns of covariate effects for these factors, we can infer that males, people with higher mood scores, people in the oncology study cohort, and observations made on days further from treatment initiation exhibit higher scores on factor 4 and lower scores on factor 2, and thus tend to have lower overnight HR.

#### 3.2.2. Associations between HR Conditional Means and Explanatory Variables

We used our methodology to estimate mean HR trajectories at specific values of the explanatory variables, combining the contributions of the factors rather than considering each factor individually as above. These results are displayed in Figure 5 and Figure 6.

It is well-established that males have lower resting HR than females, and that resting HR decreases with age [24,25,26,27]. These are consistent with Figure 5a,b for sex differences, and Figure 5c,d for age differences. Self-reported mood was a predictor of HR trajectories (Figure 5e,f), especially for patients for whom a higher mood score corresponded to an earlier and higher HR nadir, and higher tonic HR during the afternoon. Differences associated with caregiver type (partner versus non-partner) were not statistically significant (Figure 5g,h).

There was no strong statistical evidence for a difference between the BMT treatment arms, but the Oncology study cohort had a lower HR nadir (Figure 6a,b). Weekends, in comparison to weekdays, had later HR nadirs at lower HR levels (Figure 6c,d), especially for caregivers. During fall/winter, the HR nadir for patients appears later in the morning and at a lower HR (Figure 6e,f). This is plausibly related to the fact that nearly all of the patients were residing in locations with much less sunlight during fall/winter. Days since treatment initiation was associated with HR for patients, but not for caregivers. The more time that has passed since treatment initiation, the later and lower is the HR nadir for patients (Figure 6g).

In general, caregivers exhibited unimodal HR patterns while patients more often had bimodal patterns (except for younger patients with higher mood score). A shift toward later nadir times in patients was most prominently associated with older age and worse mood (Figure 5c,e). Caregivers are generally healthier compared to the corresponding patients in a dyad. We speculate that a bimodal HR pattern and later HR nadir reflects feebleness and its effect on daily routines, and may be used to differentiate more vulnerable cancer patients.

## 4. Discussion

In this study, we used a novel statistical approach and integrated data from two independent real-world studies employing wearable sensor technologies that monitored human physiology in cancer patient/caregiver dyads during HCT therapy for a total of 15,124 dyad days. To our knowledge, these are the largest dyadic datasets of continuously monitored HR and self-reported mood scores. Using 24 h HR data from 215 cancer dyads, we found statistically robust evidence for four independent factors of dyadic variation. Our investigation illustrated the dynamic nature of dyadic contributions throughout a day, with at least one out of four factors showing inter-dyadic differences at any given time of day.

Our study revealed that the dominant dyadic variation was synchronous, meaning that deviations from the mean HR trajectory tended to occur at the same time of day and had the same direction within a dyad. This interconnection in physiological states explains up to 7% of the dyadic variance. Such synchronicities between patient–caregiver physiological signals have been previously discussed, although primarily in controlled conditions over short time spans [28,29]. Our work extends these findings into real-world settings and across a much more extensive time frame.

The degree of synchronicity that we observed may be due to shared daily routines or emotional states, given the close relationship between the dyads. Our study did not acquire contextual states, such as the specific activities or stressors that occur throughout each study day. Nevertheless, the recognition of this synchronicity has the potential to transform the way we monitor patients in a dyadic context, promoting proactive engagement in healthcare and informing personalized care strategies.

In our analysis, key explanatory variables, such as age, sex, and mood significantly correlated with at least one dyadic or individual HR factor. Our findings for age and sex differences were consistent with extensive prior evidence about such demographic groups [30]. The expected finding of a later HR nadir on weekends and during the winter months further supported the validity of our data and analyses.

Intriguingly, we found that cancer patients displayed a bimodal HR pattern over a 24 h period, whereas caregivers had a unimodal HR pattern. We speculate this reflected daytime naps or periods of inactivity in the cancer patients. Of note, patients with better self-reported mood had an earlier HR nadir and were more active in the afternoon (i.e., toward a unimodal pattern). It is possible that changes in HR patterns in dyads could offer key insights into how ‘wellness’ intertwines with physiological parameters. This could inform ways to monitor and manage patients’ and caregivers’ health outside of the traditional clinical settings, potentially in real-time. We also observed that as days from treatment initiation increased, cancer patients had a lower morning HR nadir, again possibly reflecting increasing ‘wellness’ or adaptation to treatment-related stresses. Taken together, these findings suggest that a lower and/or earlier morning HR nadir may be associated with a favorable interplay between physiological and emotional well-being.

Our findings were obtained by applying a novel regularized variant of canonical correlation analysis, a classical method that has been successfully applied in numerous areas of science, e.g., [31]. The HR data considered here were high dimensional and functional, which generally required either regularization (e.g., [32,33]) or pre-smoothing [34] to balance bias and variance and obtain meaningful results. Our approach employed regularization of unsmoothed data; rather than using ridge-type regularization, we framed the regularization in terms of the trade-off between correlation and explaining variance, which naturally led to a combination of CCA and PCA objectives, yielding a single objective that can be optimized over a Stiefel manifold of orthogonal bases.

Our investigative focus was on recurrent anomalies in 24 h HR trajectories, and therefore we did not discuss transient phenomena that existed on shorter time scales, such as bouts of physical activity that do not occur at recurrent times of day. Another feature of our approach is that we directly analyzed the HR monitor data with minimal pre-processing. As a result, the factor loadings shown in Figure 3 and the predicted marginal means shown in Figure 5 and Figure 6 are much smoother than would be seen in the raw HR data. We do not aim here to fully distinguish physiologic from non-physiologic variation—while the factor parameter estimates and predicted marginal means should be robust to non-physiologic noise, the reported dyadic variance of 7% may be biased downward due to the presence of non-physiologic artifacts in the data.

Going forward, integrating a wider range of explanatory variables including contextual states and PRO data could enrich our understanding and lead to interventions that jointly target patients’ health and caregivers’ well-being. The field is well-poised to leverage low-cost, non-invasive wearable sensors in real-world settings [15,35]. This would foster personalized healthcare strategies and monitoring for cancer dyads, ultimately improving their outcomes [6].

## 5. Conclusions

We developed and applied a novel analytic method to investigate a dyadic dataset comprised 24 h HR data from cancer patients and their caregivers who self-monitored over an extended period. Our use of dimensionality reduction revealed a small collection of optimized loading patterns that represented mutually independent factors characterizing dyadic HR variation. These factors reflect patterns of variation in individual and dyadic 24 h HR levels capturing “chronotype” differences, as well as patterns of activity and rest throughout the day in this unique population of individuals going through an intensive medical intervention, along with their caregivers.

The association between better self-reported mood and a unimodal HR pattern and earlier HR nadir suggest a possible interplay between emotional and physiological well-being. These findings offer insight into the complex interplay between a physiological marker (HR) as a proxy for an emotional state (mood) in the context of cancer patients and their caregivers in real-world settings. The use of consumer-grade wearables and rigorous data analyses offered a means for examining relationships within dyadic data, revealing opportunities for research on interpersonal influences on health behaviors. A key takeaway from our study is that novel statistical analyses were applied to real-world physiological data obtained from consumer-grade wearable sensors in a vulnerable cancer dyad population. Our findings suggested that better mood was associated with a specific HR pattern, implying a potential interaction between emotional and physiological health that could be objectively measured. This work will facilitate future research and application of such analyses using wearable sensors.

## Figures and Tables

**Figure 1 bioengineering-11-00095-f001:**
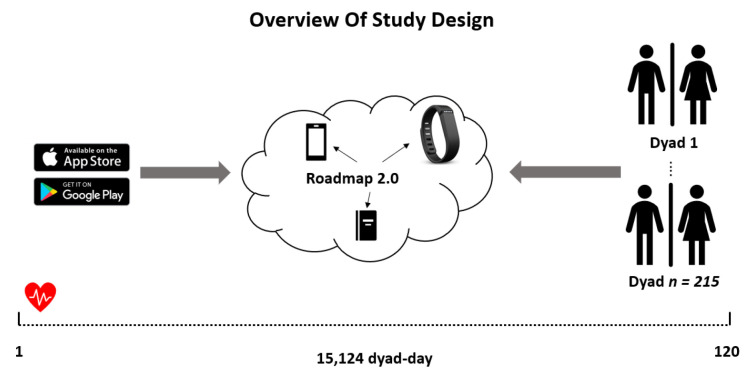
Overview of study design.

**Figure 2 bioengineering-11-00095-f002:**
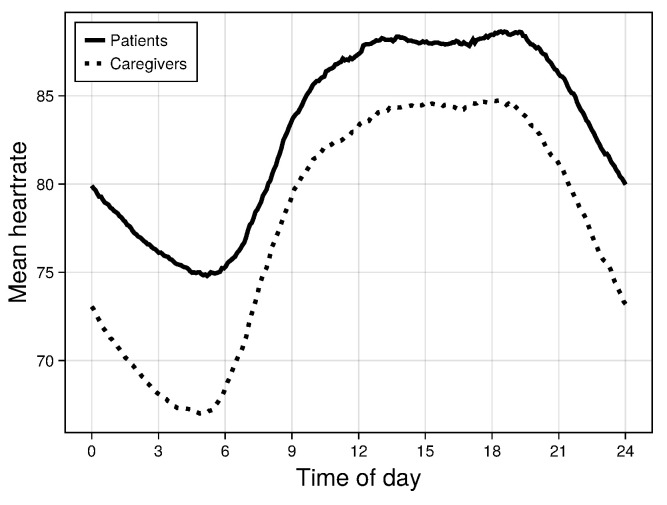
Mean HR trajectories for patients and caregivers.

**Figure 3 bioengineering-11-00095-f003:**
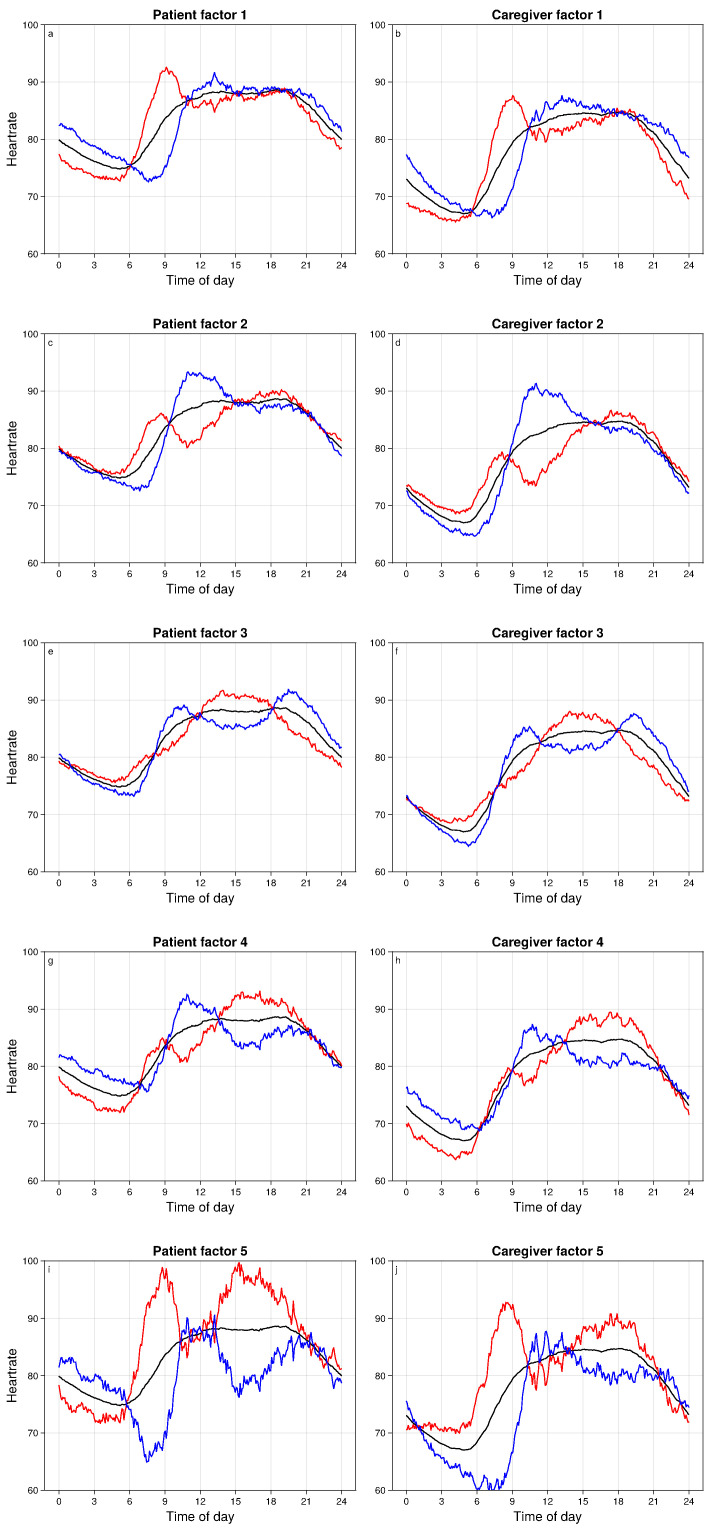
Loading patterns for five factors, for patients (**left column**) and caregivers (**right column**). The black curve is the mean, red/blue curves correspond to ±1 SD from the mean.

**Figure 4 bioengineering-11-00095-f004:**
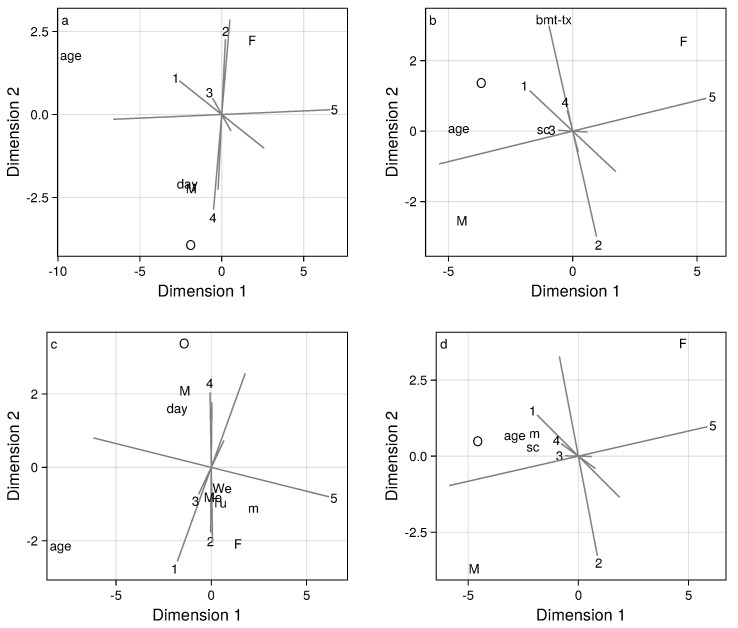
Biplots depicting regression coefficients for each factor for patients (**a**,**c**) and caregivers (**b**,**d**), for the base model (**a**,**b**) and the model including mood (**c**,**d**). Variables are abbreviated as I (intervention), F (female), M (male), m (mood), ss (season sine), sc (season cosine), and O (onc cohort). Numbers correspond to the positive end of each projected axis.

**Figure 5 bioengineering-11-00095-f005:**
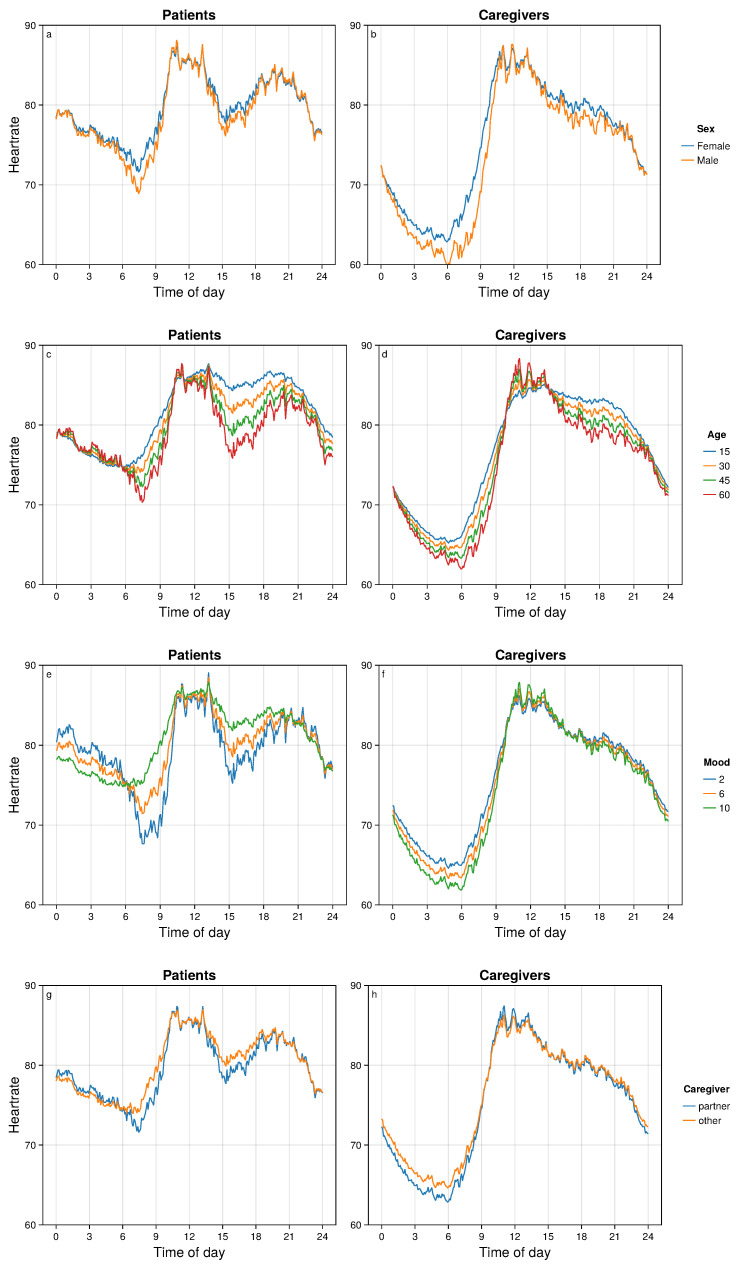
Fitted mean HRs between groups defined by sex (**a**,**b**), age (**c**,**d**), mood (**e**,**f**), and caregiver type (**g**,**h**). Patients are shown in the left column and caregivers are shown in the right column.

**Figure 6 bioengineering-11-00095-f006:**
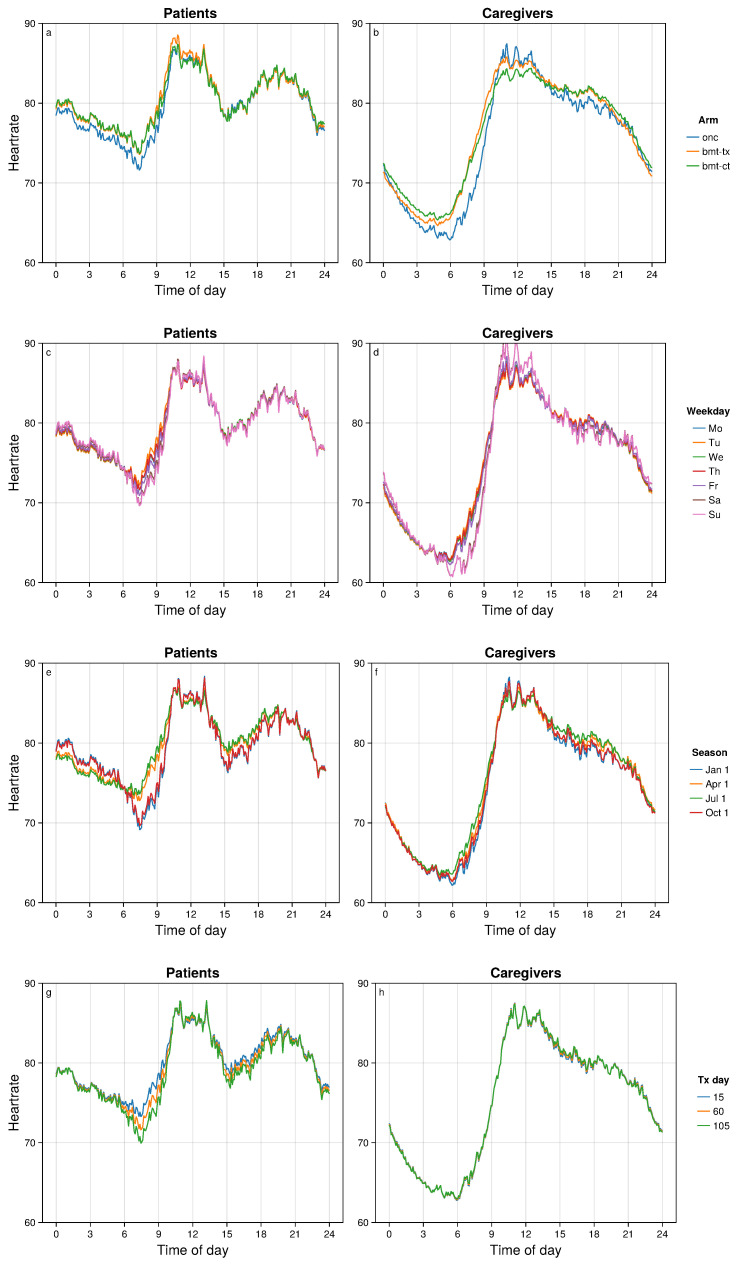
Fitted mean HRs between groups defined by cohort arm (**a**,**b**), day of week (**c**,**d**), season (**e**,**f**), and day of treatment (**g**,**h**). Patients are shown in the left column and caregivers are shown in the right column.

**Table 1 bioengineering-11-00095-t001:** Patient/caregiver correlations for each factor (*r*), with lower and upper 95% confidence bounds (LCB, UCB) from a split-sample analysis. Per-factor ${\mathrm{PVE}}_{\max}$ and cumulative proportions of explained variance (Pillai PVE) are shown for each factor.

				Patient	Patient	Caregiver	Caregiver
Factor	*r*	LCB	UCB	PVE_max_	Pillai PVE	PVE_max_	Pillai PVE
1	0.36	0.14	0.43	0.08	0.01	0.13	0.01
2	0.28	0.16	0.36	0.05	0.02	0.13	0.02
3	0.24	0.12	0.31	0.04	0.04	0.06	0.04
4	0.16	0.01	0.24	0.06	0.07	0.09	0.06
5	0.09	−0.12	0.31	0.28	0.11	0.35	0.11

**Table 2 bioengineering-11-00095-t002:** Z-scores of regression main effects for explanatory variables predicting scores of each factor in patients and in caregivers. Z-scores equal to or exceeding 2 in magnitude are bolded.

	Patient					Caregiver				
Variable	1	2	3	4	5	1	2	3	4	5
Male sex	0.7	**−2.6**	**−2.8**	**2.8**	−1.1	1.4	**2.1**	1.4	1.1	**−3.8**
age	**3.9**	1.3	**3.0**	−0.6	**−4.0**	**2.4**	−0.6	1.4	−1.6	−1.3
mood	**3.0**	**−2.2**	0.3	1.0	**3.7**	1.1	−1.8	1.0	**2.2**	−1.5
partner	0.7	−0.9	0.4	−0.5	−1.2	1.8	−1.9	0.1	1.9	−1.8
bmt-tx	0.9	**−2.0**	1.0	−0.7	0.1	1.6	−1.7	1.0	0.7	−0.1
onc	−0.3	**−2.7**	0.6	**2.9**	−0.9	0.7	−1.7	0.8	1.2	**−2.4**
Mo	**3.3**	0.5	1.3	−0.3	1.0	**2.5**	1.2	**2.0**	1.7	−0.3
Sa	**−2.2**	0.3	1.1	1.1	**−2.1**	**−5.3**	**−3.2**	1.2	−0.4	**−3.8**
Su	**−3.0**	−0.0	1.7	0.7	0.1	**−4.6**	**−3.2**	**2.4**	−0.4	**−3.2**
Th	1.8	1.1	−0.8	0.0	1.8	**2.7**	2.0	0.4	**2.0**	1.3
Tu	**4.0**	−0.0	−1.0	−0.3	**2.4**	**3.0**	**2.9**	1.3	**3.0**	1.9
We	**2.1**	1.6	0.3	1.0	**2.6**	1.6	1.0	0.7	1.6	−0.4
season (cos)	−1.5	−0.1	0.8	−0.9	−0.7	1.8	−0.5	**2.2**	0.2	−1.4
season (sin)	−2.0	1.2	−0.4	−0.7	−1.5	0.1	0.2	0.5	**−2.3**	−0.0
tx day	0.7	**−3.2**	0.1	**2.5**	**−2.4**	−1.2	0.4	1.5	1.8	−0.8

## Data Availability

Data are available upon request to the Corresponding Author: Sung Won Choi MD MS at sungchoi@med.umich.edu.

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
