# Peer review of "Dyadic and Individual Variation in 24-Hour Heart Rates of Cancer Patients and Their Caregivers"

_bioengineering, 2024, doi:10.3390/bioengineering11010095_

Round 1

Reviewer 1 Report

Comments and Suggestions for Authors

1. The methods used in Section 2.2 - 2.4  are very classic in signal processing, without any technical improvements or contributions. It is suggested that modern advanced machine learning or deep learning methods can be introduced in the paper, to identify related factors or find features underlying the collected data, etc.

2.  In Section 3, how do authors get the Fig.2 ?

3. Figs. 5 and 6 look flawless, but doubtable. More details should be given for them on how to calculate them. And, if existed, some outlier points should not be ignored.

Author Response

Comment 1: The methods used in Section 2.2 - 2.4  are very classic in signal processing, without any technical improvements or contributions. It is suggested that modern advanced machine learning or deep learning methods can be introduced in the paper, to identify related factors or find features underlying the collected data, etc.

Response 1: In our view, the main contribution of our study is the demonstration that dyadic structure can be identified in heart rate (HR) data obtained from wearables, and that it is possible to characterize associations between HR anomalies and other factors, such as self-reported mood.  This demonstrates the feasibility of using wearables to study interpersonal physiological relationships in unique populations, and to probe possible mechanisms behind these relationships.  We accomplished this using an original dataset covering over 15,000 days of observation for a unique population of cancer patients.  It was not our primary aim to develop new methodology, however the PCA-regularized CCA, including the formulation of its objective function and its non-Euclidean optimization were arguably essential for obtaining our conclusions.  Due to space limitations, we were not able to include extensive discussion of this approach.  Deep learning methods can be very powerful but it is not clear how standard deep learning methods can be used to assess dyadic, multivariate, functional correlations while also rigorously decomposing the variance into components attributable to dyadic, individual, and other sources.   Our approaches, while resembling classical methods, are adapted to work with heterogeneous high-dimensional data and are tailored to allow rigorous statistical inference for the parameters of scientific interest.

Comment 2: In Section 3, how do authors get the Fig.2 ?                                         

Response 2: Our analysis is focused on deviations from the marginal mean (where “marginal” refers to the mean over all person/days in a specific dyadic role, either patient or caregiver).  Figure 2 shows an estimate of the marginal mean trajectory for each role, using available data. That is, at each time point within the 24 hour day, we take the unweighted average over all available data (note that this is an average of thousands of observations).  Since our analysis focuses on deviations from the mean, rather than the mean itself, we only briefly discuss figure 2 in the manuscript.  The formula defining the estimated marginal mean (along with some other key formulas) is now numbered for easier reference.  We have also added some explanation to section 2.3.1 to better motivate the role of this marginal mean trajectory in the overall analysis.

Comment 3: Figs. 5 and 6 look flawless, but doubtable. More details should be given for them on how to calculate them. And, if existed, some outlier points should not be ignored.

Response 3: Figures 5 and 6 are predicted marginal means from the regularized CCA analysis.  It is true that individual HR trajectories will not look so “flawless”.  The regularized CCA analysis effectively acts as a smoother, and moreover we have thousands of person-days in each condition to form these estimates, so spurious variance and anomalies that are present in individual HR trajectories will be almost completely suppressed in the predicted marginal means.  We are not ignoring any outlier points, but due to our focus on recurrent 24-hour HR patterns, transient anomalies and outliers that are not synchronous across a substantial fraction of the population do not impact the predicted means.  We have expanded the discussion of these points in section 4 to provide more context for the reader.

Reviewer 2 Report

Comments and Suggestions for Authors

Interesting comparison of heart rate (co)variations between patients and caregivers is presented in “Dyadic and individual variation in 24-hour heart rates of cancer patients and their caregivers”. Low-cost wearables from Fitbit were used together with mHealth app called Roadmap. Results and thoroughly analyzed and nicely presented. Language usage is reasonably good. Good work and nice presentation.

Author Response

Comment: Interesting comparison of heart rate (co)variations between patients and caregivers is presented in “Dyadic and individual variation in 24-hour heart rates of cancer patients and their caregivers”. Low-cost wearables from Fitbit were used together with an mHealth app called Roadmap. Results and thoroughly analyzed and nicely presented. Language usage is reasonably good. Good work and nice presentation.

Response: We are grateful to the reviewer for their favorable review of our manuscript and encouraging feedback. Their acknowledgment of the good work and nice presentation of the analysis and results are greatly appreciated.

Reviewer 3 Report

Comments and Suggestions for Authors

The article describes a novel analytic method to investigate a dyadic dataset 355 comprised of 24-hour HR data from cancer patients and their caregivers.

The article is well written and its scientific contribution is easy to find and understand.

Few minor concerns:

- a more rigorous state of art is absolutely necessary;

- a comparative analysis of the obtained results with the proposed method vs other similar approaches is needed;

- in conclusion section further improvements of the proposed method will give a plus for the article.

Author Response

Comment 1: The article describes a novel analytic method to investigate a dyadic dataset 355 comprised of 24-hour HR data from cancer patients and their caregivers. The article is well written and its scientific contribution is easy to find and understand.

Response 1: We appreciate Reviewer 3’s thorough and careful appraisal of our manuscript and acknowledgement of its scientific contribution.

Comment 2: Few minor concerns included: (a) a comparative analysis of the obtained results with the proposed method vs other similar approaches is needed; (b) in conclusion section further improvements of the proposed method will give a plus for the article.

Response 2: We agree with Reviewer 3 that a more rigorous state of art is necessary, which is what motivated our study. Reviewer 3 also identified a few minor concerns. We appreciate and agree with the suggestion that a comparative analysis of the obtained results with the proposed method vs other similar approaches is needed but we respectfully believe it is out of the scope of the current paper’s objective. Because it is beyond the current scope, we certainly plan this for future studies that will include other approaches. Based on other reviewers’ suggestions, we have included new edits to the Conclusion section that we hope has further improved the manuscript.

Reviewer 4 Report

Comments and Suggestions for Authors

Kumar et al have applied a regularized variant of canonical correlation analysis to investigate the heart-rate daily profile in dyads composed of a cancer patient and the associated caregiver.

The study is conducted is a rigorous manner, the presentation of the methods is fairly clear, although it might be improved at some points. For instance, I deem Fig. 4 difficult to interpret and not clearly explained in the main text.

Also the presentation of the Factor identification may be improved: besides the technical description, it would be useful to give a more intuitive description.

The weaker point of the paper is the conclusions section, since these are quite brief and hazy.

Author Response

Comment 1: Kumar et al have applied a regularized variant of canonical correlation analysis to investigate the heart-rate daily profile in dyads composed of a cancer patient and the associated caregiver. The study is conducted in a rigorous manner, the presentation of the methods is fairly clear, although it might be improved at some points. For instance, I deem Fig. 4 difficult to interpret and not clearly explained in the main text.

Response 1: We thank Reviewer 4 for their review and favorable feedback. Their acknowledgment of the rigorous conduct of the study is greatly appreciated. We agree that our discussion of the biplots was not very accessible (Figure 4).  Biplots can be challenging to explain, yet we find them to be an effective way to consolidate interpretation of the simultaneous relationships between multiple factors (five in our case), and multiple predictors (up to 15).  We have therefore extended the discussion of the biplots in the revised manuscript to hopefully make it easier and more intuitive to understand the information that they convey.

Comment 2: Also the presentation of the Factor identification may be improved: besides the technical description, it would be useful to give a more intuitive description.

Response 2: We appreciate the Reviewer’s comment that the presentation of the Factor identification may be improved. We agree that identification and rotation are subtle points in any dimension reduction analysis and have added two sentences to section 2.3.3 explaining how we rotated the factor loadings to make them identified.

Comment 3: The weaker point of the paper is the conclusions section, since these are quite brief and hazy.

Response 3: We have added a few sentences to the conclusions section to explicitly state what we feel are the most important findings from our study. Thank you for bringing this to our attention. Keeping in line with the journal’s instructions of limiting the text file to a defined word limit, we attempted to succinctly summarize the intent of the paper, followed by results, and summary of the discussion. We added a line at the end of the Conclusions that hopefully brings clarity in wrapping up the conclusions of the paper. “A key takeaway from our study is that novel statistical analyses were applied to real-world physiological data obtained from consumer-grade wearable sensors in a vulnerable cancer dyad population. Our findings suggested that better mood was associated with a specific HR pattern, implying a potential interaction between emotional and physiological health that could be objectively measured. This work will facilitate future research and application of such analyses using wearable sensors.”

Reviewer 5 Report

Comments and Suggestions for Authors

General comments

The authors aimed to uncover patterns in the 24-hour heart rate variations of cancer patients and their caregivers using data from two independent datasets. Employing a novel method, researchers identified four distinct factors reflecting dyadic covariation in circadian heart rate. I found the study interesting and novel, proposing new approaches for the investigation of interpersonal physiological relationships in a quantitative way.

Specific comments

1.     How the 4 factors could be interpreted?

2. Patient/caregiver correlations for each of the 4 factors are not so high (Table 1), though the variations of HR in the examples shown in Fig 3 are quite similar. Why?

3.     How much-trusted self-reporting of mood?

4.     I suggest the numbering of all equations. All graphs must have labels and dimensions. Show standard deviations by using dash lines.

Author Response

Comment 1: The authors aimed to uncover patterns in the 24-hour heart rate variations of cancer patients and their caregivers using data from two independent datasets. Employing a novel method, researchers identified four distinct factors reflecting dyadic covariation in circadian heart rate. I found the study interesting and novel, proposing new approaches for the investigation of interpersonal physiological relationships in a quantitative way.

Response 1: We appreciate Reviewer 5’s thoughtful review and encouraging comments. Their acknowledgement that the study was interesting and novel is greatly appreciated.

Comment 2: How the 4 factors could be interpreted?

Response 2: HR variation in humans is complex and multifaceted, and this is likely exaggerated in patients undergoing a complex medical treatment and their caregivers.  A natural starting point for the interpretation of our findings is that people exhibit different “HR chronotypes”, reflecting when and how rapidly people arise in the morning and conversely when they go to sleep in the evenings.  Although this study focuses on HR not sleep per-se, it seems very likely that the dominant determinant of the 24 hour HR cycle is the diurnal sleep/wake cycle.  We note that the morning and evening components of the chronotype are naturally coupled (people who rise earlier will tend to sleep earlier), and the factor loading patterns reflect this coupling.  Beyond this, we find less-anticipated patterns of HR anomalies including the presence of either one or two local HR maxima in the afternoon, and variation in the minimum HR overnight.  We prefer not to speculate too much in the paper but it seems likely that our subjects who are generally not working or otherwise forced to adhere to a strict schedule may choose to variously rest or participate in activities during the afternoon, and may variously choose to follow a normative three-meal schedule or may eat fewer meals at nontraditional times.  Moreover some may have trouble sleeping overnight or take long naps during the day.

Comment 3: Patient/caregiver correlations for each of the 4 factors are not so high (Table 1), though the variations of HR in the examples shown in Fig 3 are quite similar. Why?

Response 3: We may expect to find strong correlations between socially-connected individuals in an experimental setting where, for example, the two members of a dyad are collaborating on a task or intensively interacting with each other.  However, our data reflect naturalistic 24-hour behavior over many weeks.  We expect that during much of the day each member of a dyad is primarily engaged in individual activities, and moreover these are distinct individuals with unique physiology and often at different life stages.  For this reason, dyadic correlations of around 7%, as we found, may capture most of the true interdependence that is reflected in the HR.  However, as noted in section 4 (and we have expanded this discussion in the revision), we acknowledge that some fraction of the HR variance captured by wearables may be non-physiologic.  In this study we chose to work with unselected data and hence any non-physiologic artifacts in the wearable data may attenuate the dyadic component of the variance.

Comment 3:  How much-trusted self-reporting of mood?

Response 3: The Reviewer raises a relevant point. This is a well-recognized limitation in self-reported data, which is considered subjective in nature. However, in medicine, it remains an important source of information and data (i.e., obtaining the voice and perspective of the patient). Nonetheless, it is our research program’s goal to identify objective markers of conditions or disease, which may include physiological markers, such as HR. In this paper, we sought out to include both variables in our models and indeed identified correlations between mood and HR.

Comment 4: I suggest the numbering of all equations. All graphs must have labels and dimensions. Show standard deviations by using dash lines.

Response 4: Reviewer 5 raises a relevant point. We followed the template provided by Overleaf/LaTex and have now enumerated all of the equations, as suggested.

Round 2

Reviewer 1 Report

Comments and Suggestions for Authors

Authors have addressed all my comments.